# Modified Hyaluronic Acid-Laminin-Hydrogel as Luminal Filler for Clinically Approved Hollow Nerve Guides in a Rat Critical Defect Size Model

**DOI:** 10.3390/ijms22126554

**Published:** 2021-06-18

**Authors:** Zhong Huang, Svenja Kankowski, Ella Ertekin, Mara Almog, Zvi Nevo, Shimon Rochkind, Kirsten Haastert-Talini

**Affiliations:** 1Institute of Neuroanatomy and Cell Biology, Hannover Medical School, 30623 Hannover, Germany; huang.zhong@mh-hannover.de (Z.H.); kankowski.svenja@mh-hannover.de (S.K.); ella.i.ertekin@stud.mh-hannover.de (E.E.); 2Center for Systems Neuroscience (ZSN) Hannover, 30559 Hannover, Germany; 3Research Center for Nerve Reconstruction, Department of Neurosurgery, Tel-Aviv Sourasky Medical Center, Tel Aviv University, Tel Aviv 69978, Israel; maraa@tlvmc.gov.il (M.A.); shimonr@tlvmc.gov.il (S.R.); 4Department of Human Molecular Genetics and Biochemistry, Sackler School of Medicine, Tel Aviv University, Tel Aviv 69978, Israel

**Keywords:** sciatic nerve regeneration, nerve guidance conduit, collagen, chitosan, hydrogel luminal filler

## Abstract

Hollow nerve guidance conduits are approved for clinical use for defect lengths of up to 3 cm. This is because also in pre-clinical evaluation they are less effective in the support of nerve regeneration over critical defect lengths. Hydrogel luminal fillers are thought to improve the regeneration outcome by providing an optimized matrix inside bioartificial nerve grafts. We evaluated here a modified hyaluronic acid-laminin-hydrogel (M-HAL) as luminal filler for two clinically approved hollow nerve guides. Collagen-based and chitosan-based nerve guides were filled with M-HAL in two different concentrations and the regeneration outcome comprehensively studied in the acute repair rat sciatic nerve 15 mm critical defect size model. Autologous nerve graft (ANG) repair served as gold-standard control. At 120 days post-surgery, all ANG rats demonstrated electrodiagnostically detectable motor recovery. Both concentrations of the hydrogel luminal filler induced improved regeneration outcome over empty nerve guides. However, neither combination with collagen- nor chitosan-based nerve guides resulted in functional recovery comparable to the ANG repair. In contrast to our previous studies, we demonstrate here that M-HAL slightly improved the overall performance of either empty nerve guide type in the critical defect size model.

## 1. Introduction

Peripheral nerve injures (PNIs) arising from trauma with a 2–3% incidence have a strong impact on patients due to their accompanying sensory and motor function loss; for example, dysesthesia, paralysis, and neuropathic pain [1]. Although peripheral nerves have an intrinsic potential for axonal and functional regeneration after injury, the overall outcome is often poor and varies according to patients’ age and severity, and location of the injury. Despite numerous efforts in pre-clinical and clinical research, the autologous nerve graft (ANG) still represents the gold standard reconstructive approach for longer nerve gaps [2], despite about 20% of patients not regaining useful function after this type of treatment [3]. Additionally, ANG repair is affiliated to some shortcoming, such as donor-site morbidity, including possible development of neuropathic pain, prolonged surgery times, and limited availability of grafting material for extended lesions [3,4]. 

Over the last decades, numerous bioartificial nerve grafting approaches have been designed and pre-clinically studied [5,6]. As a result, hollow nerve guidance conduits based on natural as well as some synthetic materials were approved for clinical use as bridging material for nerve gap lengths of up to 3 cm [3,5]. The most widely used tubular graft is probably the collagen-based NeuraGen^®^ Nerve Guide for which a large body of pre-clinical and clinical data demonstrate its value as an off-the-shelf product [3,7]. The probably most recently approved nerve regrowth tube for clinical use is Reaxon^®^ Nerve Guides [8]. This tubular graft is made from the natural polysaccharide chitosan and we have, among others, contributed to its comprehensive pre-clinical evaluation [9,10,11,12,13,14]. 

Nerve guidance conduits are more widely used for digital nerve repair [7,8], while repair with ANG is more successful when applied across larger nerve gaps. Therefore, the clinical use of hollow nerve guidance conduits for extended nerve gap reconstruction is still not favored above the gold standard [8,15]. It is conclusive that researchers and clinicians seek for an option of improving the performance of nerve guidance conduits. Combinations of bioartificial tubular grafts with growth-enhancing substrates such as specific neurotrophic factors, regeneration supporting cells, or luminal additives or fillers composed of extracellular matrix (ECM) components are frequently considered to be promising [9,15,16,17]. A three-dimensional ECM is naturally formed within the nerve defect and Schwann cells of the repair phenotype contribute to its assembly [18,19]. The ECM plays an important role during regeneration and promotes axonal regrowth by providing a growth-permissive structure and by enriching molecules that can activate regeneration-associated signaling pathways [15,20]. A major class of ECM components are laminins, which have been shown to support differentiation and directed migration of Schwann cells, axon myelination during the processes of peripheral nerve regeneration [21,22,23]. Hyaluronic acid (HA), another key component of the ECM, plays an important role in neural proliferation, differentiation, migration, survival, and cell signaling [24]. As Schwann cells represent the glia cells of the peripheral nervous system and are as such crucially involved in maintenance and regeneration processes [19], both, laminin and HA have been studied and demonstrated their potential as nanomedicals for supporting cellular regeneration [25]. How extracellular matrix components can work as instructive engineered microenvironments for peripheral nerve repair approaches has recently been reviewed in [26]. Here the authors also highlight the good potential for clinical use of these molecules due to their well standardized and reproducible synthesis [26]. 

In our previous work, we have already evaluated different hydrogel formulations as luminal fillers for either collagen-based NeuraGen^®^ Nerve Guides [27,28] or chitosan-based Reaxon^®^ Nerve Guides [29,30]. Recently we evaluated a 0.2% hyaluronic acid-laminin-hydrogel (HAL) as luminal filler of chitosan-based nerve guides in the acute repair of 15 mm critical length sciatic nerve defects in the adult rat [30]. Our results indicated that the previous formula of the HAL was not yet optimal for robust support of axonal regeneration in the immediate repair approach. Interestingly, a further modified formulation of the gel with increased concentration introduced into collagen-based nerve guides did well support axonal regeneration in a rabbit model of delayed repair of 25 mm tibial nerve defects [28]. This finding contributes to the view that complex structural support within nerve guidance conduits is a valuable condition for increasing functional outcome after nerve gap repair [26].

In the current study, we primarily aimed in detecting any material interference between a modified HAL (M-HAL) and the two different types of nerve conduits, used in our studies so far. Therefore, we introduced M-HAL in two different higher concentrations of hyaluronic acid into either collagen- or chitosan-based nerve guides and comprehensively evaluated the outcome of functional and axonal regeneration in our challenging immediate repair rat model. Over a period of 120 days, we compared four experimental groups with hydrogel filled nerve guides to one group in which we applied the gold standard, reversed autograft repair (ANG), and one group in which we used the empty collagen-based graft. We further compared our new data with our own previously published data on the use of empty chitosan-based nerve guides in the same model [30,31]. Also in comparison to our previous work evaluating HAL [30], we performed additional gene expression profiling in vitro in Schwann cells cultured within M-HAL. 

We present here data derived from a challenging pre-clinical approach of acute nerve repair, confirming that M-HAL performs equally when introduced in either clinically approved nerve guide. Our data further indicate that the performance of the empty tubular grafts could be increased to some small degree with the addition of M-HAL in either of the tested concentrations. From our additional in vitro testing, we conclude that this effect is more likely attributed to a structural support of axonal outgrowth than to M-HAL, providing a supportive milieu for maintaining the repair phenotype of invading Schwan cells. 

## 2. Results

For clarity of the presented results, we shortly present our experimental design in Table 1.

### 2.1. Functional Assessment of Motor Recovery: Repeated Noninvasive Electrodiagnostic Recordings

We used noninvasive electrodiagnostic recordings for monitoring functional motor recovery during the observation period after immediate repair of the 15 mm sciatic nerve defect in adult rats. The recordings were repeated every 30 days from 60 days after surgery onwards until 120 days. Evocable compound muscle action potential (CMAP) amplitude areas, amplitude ratio, and axon loss, were subsequently calculated from the recorded CMAPs of the anterior tibial (TA, Table 2) and the plantar (PL) muscles (Table 3). The CMAPs of the contralateral, healthy sciatic nerve of the respective animal, recorded at the very same day, served as reference signal. 

We first report the recovery rates for all examined treatments; they indicate the percentage of animals per group from which evocable CMAPs could be recorded (“Recovery rates” column in Table 2 and Table 3). At 60 days after reconstruction of the sciatic nerve, the ANG-treated animals were the first group to display a 100% recovery rate (percentage of animals per group showing evocable CMAPs) of the TA muscle (Table 2). Likewise, a first animal provided with CNG + 0.4% M-HAL showed evocable CMAPs in the TA muscle. No signs of motor recovery were detectable in the other animals, provided with different combinations of bioartificial nerve grafts and M-HAL luminal filler. Please note that two animals died due to anesthesia complications after the first functional evaluation (ANG-group and CNG + 0.4% M-HAL-group).

At 90 days after surgery, the rate of functional motor recovery observed in the ANG group (100%) was not changed. Compared to the first recording, however, the functional recovery rate in CNG + 0.4% M-HAL-treated animals improved to 60% (Table 2). First signs of functional motor recovery of the TA muscle were also detectable in the NNG + 0.4% M-HAL (50%), CNG + 0.7% M-HAL (50%), and NNG-treated (16.7%) groups. Still, no CMAPs were evocable from NNG + 0.4% M-HAL-group animals. 

At the endpoint of the observation period, 120 days post-surgery, ANG and NNG + 0.7% M-HAL-treated animals demonstrated a 100% functional motor recovery rate for the TA muscle (Table 2). Partial motor recovery rates were detected after nerve reconstruction with CNG + 0.4% M-HAL (80%), and NNG (50%). The recovery rate in the CNG + 0.7% M-HAL group (50%) was not further changed compared to 90 days post-surgery. Remarkably, the TA muscle functional recovery rate observed in the NNG + 0.4% M-HAL group was increased from 0 to 66.7% (compared to 90 days post-surgery). 

Overall, in comparison to the more proximal TA muscle, electrodiagnostic recordings of PL muscles at the foot sole, revealed lower rates of functional motor recovery (Table 3 “Recovery rates” column). The ANG treatment, exclusively, achieved full functional recovery (100% recovery rate) after 90 days after displaying 66.7% recovery rate at 60 days post-surgery. No functional PL muscle motor recovery was observed in animals treated with different combinations of artificial nerve grafts and luminal filler 60 and 90 days post-surgery (Table 3). At 120 days after nerve reconstruction, partial recovery of PL muscle motor function was detected in the following groups: CNG + 0.7% M-HAL (33.3%), NNG (16.7%), and NNG + 0.7% M-HAL (16.7%). It is noteworthy, that out of the only three CNG + 0.7% M-HAL-treated animals demonstrating reinnervation of the TA muscle (50% recovery rate at 120 days), two also showed reinnervation of the PL muscle.

We further analyzed the recorded CMAPs for quantitative parameters, e.g., their CMAP amplitude area (AUC), from which we also calculated the percentile loss of functional axons in comparison to the contralateral non-lesioned nerve, and their CMAP amplitude ratio, displaying the degree of recovery in relation to the non-lesioned side. The amplitude area of the recorded CMAPs reflects the number of functional axons reinnervating the target tissue, and can therefore be regarded as a sign of nerve recovery. As a reference, the CMAP AUC of the TA (*n* = 22) and PL muscles (*n* = 12) on the non-lesioned (contralateral) side were calculated as mean ± SEM: TA CMAP AUC = 31.44 ± 1.70 [ms * mV]; PL CMAP AUC = 2.11 ± 0.28 [ms * mV].

The application of ANG was superior to all other groups with regard to the CMAP amplitude area recorded from the TA muscle at all recording time points (Table 2 “CMAP amplitude area” column). The ANG group reached almost non-lesioned nerve reference values at 90 days and 120 days post-surgery. Even at the 120 days endpoint of the observation period, the animals treated with bioartificial nerve grafts and M-HAL showed significantly lower TA muscle mean CMAP AUC compared to ANG-treated animals at 60 days post-surgery. No significant differences among groups provided with different combinations of bioartificial nerve grafts and M-HAL could be observed. 

Similar results were obtained by comparing the AUC of the recorded PL-CMAPs (Table 3 “CMAP amplitude area” column). It is noteworthy, however, that the mean values calculated in the ANG group were still much lower compared to the respective non-lesioned nerve reference values at all evaluation time points until 120 days post-surgery. 

The loss of functional axons (Table 2 and Table 3 “axon loss” column and Figure 1A) and the inversely correlating ratio of the respective CMAP amplitude (Table 2 and Table 3 “CMAP amplitude ratio” column), underscore the above described results regarding functional motor recovery. ANG-treated animals clearly recovered during the observation period, showing a significantly lower axon loss (Figure 1A) and higher amplitude ratio when compared to the groups treated with bioartificial nerve guides at the same time points. No significant differences were observed among the animals treated with different combinations of bioartificial nerve guides and M-HAL. It is obvious, however, that over time CMAP amplitude ratios recorded from the TA muscle increased in all investigated groups, displaying a progress in functional recovery. This progress was also significant not only for ANG treatment but also in the NNG + 0.7% M-HAL group, in which functional recovery was clearly progressed more than in all other groups treated with bioartificial nerve guides and M-HAL (Table 2).

For ethical and animal welfare reasons (reduction of animal numbers used), we have not again investigated in the current study, experimental groups we have already been investigating before in the same model (15 mm rat sciatic nerve gap, animals from the same breeder, strain, sex, and matched age). However, for a more comprehensive evaluation of the new data derived from the current study, we want to report specific previously published data again [30,31]. These data relate to the use of empty chitosan-based nerve guides (CNG) in Stößel et al. 2018 [31], and to the use of chitosan-based nerve guides filled with hyaluronic acid (HA) or with the non-modified HAL both in a concentration of 0.2% in Dietzmeyer et al. 2020 [30]. 

With regard to recovery rates, in both previous studies and in accordance to the current study, the ANG repair supported recovery of evocable TA muscle CMAPs in all animals (100%) from 60 days post-surgery onward [30,31]. Reconstruction of the 15 mm sciatic nerve gap with empty CNG resulted in a TA muscle recovery rate of 0% at 60 days, 37.5% (in 3 out of 8 animals) at 90 days, and 42.9% (in 3/7 animals) at 120 days post-surgery [31]. The endpoint recovery rate with empty CNGs was therefore slightly lower than for empty collagen-based NNG (50%) in the current study. Using CNG filled with 0.2% HA for nerve gap reconstruction even reduced the TA muscle recovery rates to 0% at 60 and 90 days, and 20% (in 1/5 animals) at 120 days post-surgery [30]. Treatment with CNG + 0.2% HAL slightly increased and accelerated TA muscle recovery rates to 25% (in 2/8 animals) from 60 days post-surgery onward [30]. Endpoint recovery rates for CNG + 0.2% HA and CNG + 0.2% HAL treatments were therefore below those of empty CNG treatment, but also below TA muscle recovery rates of all treatments evaluated in the current study.

Accordingly, data from axon loss calculation demonstrate the same differences as described before for axons innervating the TA muscle. The results are depicted in Figure 1. Mean axon loss after ANG treatment in both preceding studies range at about 50% at 60 days post-surgery and at about 0% at 90 and 120 days post-surgery (Figure 1B,C) [30,31]. Mean TA muscle axon loss after treatment with empty CNG (Figure 1B) was 100% at 60 days, and lowered to ~70% at 90 days and ~60% at 120 days post-surgery [31]. This, in contrast to the recovery rate, displays a better support of regeneration in comparison to most of the treatments evaluated in the current study (Table 2, Figure 1A), where solely NNG + 0.7% M-HAL showed a similar result. Mean TA muscle axon loss after treatment with CNG + 0.2% HA (100% at 60 and 90 days, ~90% at 120 days, Figure 1C) and CNG + 0.2% HAL (~90% at 60 days, ~75% at 90 days, and ~70% at 120 days post-surgery, Figure 1C) [30], also recovered to a slightly better degree than after most treatments in the current study (Table 2, Figure 1A). Again, with the exception of NNG + 0.7% M-HAL treatment, displaying a mean TA muscle axon loss of ~60% at 120 days post-surgery.

With regard to recovery of PL muscle function, recovery rates after ANG treatment displayed a similar outcome in both previous studies [30,31]. Recovery rates were superior in comparison to any bioartificial nerve graft treatment in the current study (Table 3) after bridging the 15 nerve gap with empty CNG (90 days: 37.5%, in 3/8 animals; 120 days: 28.6%, in 2/7 animals), also the mean axon loss ranged at slightly superior values of ~80–90% [31]. Treatment with CNG + 0.2% HA did not result in any detectable recovery of PL muscle function [30]. Treatment with CNG + 0.2% HAL, however, was slightly superior with regard to PL muscle recovery rate (25%, in 2/8 animals, [30]) to most of the bioartificial nerve graft + M-HAL treatments in the current study (Table 3). One single exception here is the treatment with CNG + 0.7% M-HAL (33.3% recovery rate, Table 3). With regard to the more quantitative parameters, however, CNG + 0.2% HAL treatment from our previous study (PL CMAP AUC: 0.42 ± 0.28 [ms * mV]; PL muscle axon loss: ~ 70%; PL CMAP amplitude ratio: 0.12 ± 0.08) [30] ranges superior to the data obtained in the current study (Table 3).

### 2.2. Functional Assessment of Motor Recovery: Muscle Weight Ratios (MWRs)

The assessment of functional motor recovery by noninvasive electrodiagnostic recordings was supported by the determination of the respective muscle weight ratios (MWRs) of the largest lower hind limb muscles. A high MWR, approximating 1, indicates a recovery in muscle mass due to successful target muscle reinnervation. Therefore, the anterior tibial (TA) muscle and gastrocnemius (GC) muscle were harvested at the 120 days endpoint of the observation period. The ratio between the muscle weights of the treated (ipsilateral) limb in comparison to the non-lesioned (contralateral) limb was calculated. Please note that the muscle weight of two animals was accidentally not recorded and that two animals died during the course of the electrodiagnostic measurements at 60 days post-surgery. Despite this, all animals reaching the endpoint of the observation period, were included in this analysis: ANG: *n* = 5; NNG: *n* = 6; NNG + 0.4% M-HAL: *n* = 5; NNG + 0.7% M-HAL: *n* = 6; CNG + 0.4% M-HAL: *n* = 5; CNG + 0.7% M-HAL: *n* = 5.

Results from the evaluation of MWRs are graphically depicted in Figure 2. The obtained data are in accordance to the findings from the electrodiagnostic measurements. Critical gap length nerve reconstruction with ANG was significantly superior to the application of all other bioartificial nerve grafts. There were no significant differences observed between groups of animals, provided with different combinations of bioartificial nerve grafts and M-HAL.

### 2.3. Evaluation of Nerve Histomorphometry

Non-lesioned, contralateral nerve segments, and segments dissected distal from the graft of the reconstructed sciatic nerve were harvested after 120 days, at termination of the observation period. The segments were, fixed, stained, embedded, and sliced into semi-thin cross-sections for histomorphometrical analysis. Figure 3 shows representative photomicrographs of cross-sectional details from the different groups. 

Stereological analysis was only performed, if the fiber density of the section was high enough to ensure a valid analysis with the described method (see section Nerve Histomorphometry). Consequently the number of analyzed samples did not for all groups match the number of evaluated animals.

Figure 4 shows a graphical summary of the results from the stereological evaluation. ANG treatment was clearly superior to all experimental groups, with regard to the cross-sectional area of the distal nerve segment (Figure 4A) and the nerve fiber density (Figure 4B) of myelinated axons within the analyzed semi-thin cross-sections. The difference in the cross-sectional area of distal nerve segments harvested from animals treated with (ANG) was significant in comparison to samples from the NNG + 0.4% M-HAL group. 

Since the cross-sectional areas of the samples are not well comparable with one another, the nerve fiber density is considered in the following. As depicted in Figure 4B, significant differences with regard to the nerve fiber density could be observed between ANG-treated animals and all other experimental groups, treated with bioartificial grafts without or with M-HAL. Mean values obtained from samples from animals treated with different combinations of bioartificial nerve guides and M-HAL did not significantly differ from each other. It is visible, however, that most of the samples obtained from nerves after M-HAL application within the nerve guides display higher nerve fiber densities than cross-sections from NNG-treated nerves. 

Figure 5 depicts results from the morphometric analysis of distal nerve segments. Significant differences between ANG samples and samples from certain other reconstruction approaches were detected for axon diameter (Figure 5A), fiber diameter (Figure 5B), and *g*-ratio (Figure 5D), while no difference was detected with regard to the myelin thickness of the regenerated myelinated axons (Figure 5C). Significant differences were detected between treatment with ANG and NNG + 0.4% M-HAL for all three previously named parameters. The mean axon diameter as well as the mean fiber diameter of regenerated myelinated axons was significantly lower in comparison to ANG-treated nerves for CNG + 0.7% M-HAL treated nerves. For the mean fiber diameters in samples from the NNG- and CNG + 0.4% M-HAL-group, no statistically significant difference to samples from the ANG-group was detected. The significantly reduced *g*-ratio of the NNG + 0.4% M-HAL-regenerated axons compared to the ANG-regenerated axons indicates that the myelin thickness of axons in the NNG + 0.4% M-HAL-samples was increased in relation to the axon diameter. No clear difference was detectable between samples obtained from the animal groups treated with different combinations of bioartificial nerve guides and M-HAL. 

In Figure 4 and Figure 5, the values derived from the single animals are indicated by dots or triangles. The latter represent values from animals showing recovery of evocable CMAPs in the PL muscle (Table 3). It is clearly visible that motor functional recovery also of the PL muscle did only occur from a certain level of nerve fiber density onward (Figure 4B). On the other hand, it is also visible that not necessarily nerves with a higher mean fiber diameter have achieved reinnervation of PL muscles within the 120 days observation time (Figure 5B).

For a closer correlation of functional motor recovery to regeneration of myelinated axons in the nerve segments directly distal to the grafts, we further analyzed the percentile fiber diameter distribution among the morphometrically analyzed axons. For a more detailed graphical presentation, the reader is kindly referred to Appendix A. While in the non-lesioned control nerve, a peak (highest % of fibers with a certain diameter) can be detected at 8–9 µm, a clear left shift is detectable for all regenerated nerve samples, including ANG (peak at 4–5 µm fiber diameter, Appendix A). The left shift is even more pronounced when bioartificial nerve grafts are used. Interestingly, myelinated axons regenerated through empty NNG or gold-standard ANG grafts displayed rather similar percentile nerve fiber diameter distributions (Appendix A). 

As already mentioned above (see Section 2.1), the current study relates in some aspects to previously published findings of our working group [30,31] that were obtained in the same model (15 mm rat sciatic nerve gap, animals from the same breeder, strain, sex and matched age). Stößel et al. 2018, amongst other treatments, evaluated the performance of empty CNG [31]. The other study of Dietzmeyer et al. 2020 moreover, investigated the regenerative properties of HA and HAL, as luminal filler of CNG [30]. In order to reduce the number of laboratory rats used in the current study, we declined to repeat these experiments. Appendix A shows that a rather similar percentile fiber distributions can be seen for myelinated axons that have regenerated through ANG or empty CNG [31] or CNG grafts supplemented with a lower concentrated hydrogel (0.2% HA or 0.2% HAL) [30].

For a more compacted view on the data, Figure 6 compiles the distribution of nerve fiber diameters of the current study (Figure 6A) and the results from re-analyzing specific groups from the study published as Stößel et al. 2018 [31] (Figure 6B), and Dietzmeyer et al. 2020 [30] (Figure 6C). The percentile quantity of fibers with a diameter below 4 µm is considerably smaller in non-lesioned control nerves (4.2–15.8%) compared to all other experimental conditions (64.2–92.8%). Furthermore, ANG-treated animals show fewer fibers with a diameter below 4 µm (64.2–75.9%) compared to animals provided with bioartificial grafts with and without luminal filler M-HAL (89.3–92.8%). Among the sciatic nerves reconstructed with bioartificial grafts, hollow grafts result in a lower quantity of fibers with a diameter ≤4 µm (CNG: 83.3%; NNG 85.2%) compared to grafts filled with 0.2% HA (90.0%, only CNG), 0.4% M-HAL (90.7–92.6%, CNG and NNG), and 0.7% M-HAL (89.3–92.8%, CNG and NNG). Interestingly, CNG grafts filled with 0.2% HAL resulted in regeneration of a rather low 79.4% of fibers with a diameter <4 µm (Figure 6C), which is similar to what could be detected in ANG-reconstructed nerves (Figure 6A–C). 

### 2.4. In Vitro Analysis of Neurotrophic Factor Expression

In accordance to our previous study, in which we additionally evaluated HAL as potential cell carrier system for transplanted Schwann cells [30], we analyzed the performance of M-HAL in vitro with regard to the question, if M-HAL affects the regeneration supportive properties of Schwann cells (SCs) cultured within the hydrogel. Therefore, primary neonatal SCs cultured in regular medium were compared to SC cultures grown in 0.2% HA, 0.4% HA, or 0.4% M-HAL. For the purposes of this study, the expression of genes encoding for the neurotrophic factors BDNF, GDNF, NGF, FGF-2, as well as the transcription factors C-jun, and Sox 2, and the Schwann cell specific calcium-binding protein S100ß was analyzed with qRT-PCR. 

Representative photomicrographs (Figure 7) taken from the different experimental culture conditions show a slightly reduced cell density for the 0.4% M-HAL condition in comparison to the other conditions at DIV1. No significant differences, however, in SC morphology and SC cell density were detectable anymore after three days in vitro. This indicates that proliferation rates of cultured SCs did quickly recover in the 0.4% M-HAL conditions. 

However, mRNA amounts were equalized before performing qRT-PCR analysis for evaluating gene expression in the cultured SCs. The determined gene expression values for neurotrophic and transcription factors are depicted in Figure 8. While no significant changes in expression of *bdnf* (Figure 8A) was detectable, gene expression levels for *gdnf* (Figure 8B), *fgf-2* (Figure 8C), and *ngf* (Figure 8D) were significantly decreased in the presence of 4% M-HAL compared to the control condition (K+). Furthermore, the *ngf* gene expression was also significantly reduced in the 0.4% M-HAL condition in comparison to the 0.2% HA- and 0.4% HA condition. Furthermore, no significant differences were detected for gene expression levels of the transcriptions factors *sox-2* (Figure 8E) and *c-jun* (Figure 8F) as well as the SC specific gene *s100β* (Figure 8G). It is, however, obvious from Figure 8E–F that the respective expression levels were lower in the 0.4% M-HAL conditions than in all other culture conditions. 

## 3. Discussion

In the current study we comprehensively evaluated, with functional and histological methods, the outcome of the reconstruction of critical gap length rat sciatic nerve defects. Nerve defects were bridged with two different nerve guides approved for clinical use. Both nerve guides, produced from the natural polymers collagen or chitosan, were filled with a modified hyaluronic acid-laminin hydrogel (M-HAL) in two different concentrations. The rationale for adding hydrogel luminal fillers into otherwise hollow nerve guidance channels is based on the fact that empty nerve conduits provide sufficient clinical outcome only when applied for short nerve gap repairs [3]. Both, laminin and hyaluronic acid (HA), are extracellular matrix components present within peripheral nerves and crucially involved in providing a regeneration supporting 3D environment for axonal regeneration [20,32,33]. Results from our own previous work [27,28,30] have driven us to hypothesize that M-HAL would improve nerve functional and axonal regeneration in acute repair of critical nerve defect lesions in the rat. We further aimed to investigate if M-HAL can provide its action independently of the nerve guidance conduits used as container for the hydrogel.

Therefore, we chose two clinically approved nerve guidance conduits, NeuraGen^®^ Nerve Guide [3] and Reaxon^®^ Nerve Guide [8]. We have been working with both of them separately in our previous work in rats [27,28,30]. Reconstruction with the different nerve guides and M-HAL combinations was compared to the gold standard procedure, nerve autografting (ANG). We have further used in the current study empty NeuraGen^®^ Nerve Guides (NNG) as control and compared all results from the current study with more control conditions, which we have recently evaluated in the same model. Repetition of these control conditions were avoided for ethical reasons. These conditions were likewise ANG repair in both previous studies [30,31], and empty Reaxon^®^ Nerve Guides (emptyCNG) [31], or CNG filled with HA or non-modified HAL, both applied in the lower concentration of 0.2% [30].

Electrodiagnostic recordings of evocable compound action muscle potentials (CMAPs) from muscles at more proximal and more distal locations distal to the lesion site give indications about the temporal progress of functional motor recovery [34]. Here, we recorded from the tibialis anterior (TA) muscle located at the lower hind limb, as well as from the plantar (PL) muscle at the foot sole. The results of the current study are, with regard to ANG repair, consistent with results of our previous work [11,12,30,31]. ANG did, once again, prove to be a reliable gold standard in the rat sciatic nerve repair model. This can be seen from the recovery rate of TA muscle (recovery rate 100% of animals showing evocable TA muscle CMAP) motor function at 60 days after the surgery and the recovery rate of PL muscle (100% recovery rate) motor function at 90 days after surgery (Table 2 and Table 3, “Recovery rates” column). Compared with the ANG-treated animals, in all other experimental groups, treated with bioartificial nerve grafts (NNG; NNG + 0.4% M-HAL; NNG + 0.7% M-HAL; CNG + 0.4% M-HAL and CNG + 0.7% M-HAL), the progress of motor recovery was slower in time and remained incomplete until the end of the observation period at 120 days after surgery. With regard to the timing of recovery, it can be summarized that animals from the ANG-group were the first demonstrating detectable recovery of TA muscle function. This was followed by a lower number of animals in the CNG + 0.4% M-HAL-group. During the follow-up measurements, animals from the NNG + 0.4% M-HAL group, however, almost caught up in number, while animals of the NNG + 0.7% M-HAL group even overruled their performance with regard to the degree of functionality. The latter was evaluated by the CMAP amplitude ratio (and correlated functional axon loss) derived from the recorded signals in comparison to the respective signals obtained from the contralateral non-lesioned nerve and TA muscle (Table 2, “CMAP amplitude” and “Axon Loss” columns). Over time, quantitative parameters for TA muscle function recovered to some degree in all investigated groups. The progress was, however, significant only for ANG treatment and, although to a significantly lower level, for nerve reconstruction with NNG + 0.7% M-HAL.

From analyzing the PL muscle function (Table 3), it can be assumed that full recovery was already achieved for ANG treatment at 120 days post-surgery, but was likely still under way for repair with any of the bioartificial grafts evaluated. It is not clear, however, if a similar progress of recovery rates and functionality values as reported for the TA muscle would have taken place for the PL muscle if the observation time would have been elongated.

In comparison to our previous studies, we can summarize that reconstruction of the 15 mm sciatic nerve gap with empty collagen-based (NNG, the current study) demonstrated a slightly better support than repair with empty chitosan-based (CNG, [31]) nerve guidance conduits. Adding cell-free 0.2% HA or 0.2% HAL [30] did clearly diminish recovery rates, while adding 0.4% or 0.7% M-HAL as luminal filler (current study) showed a trend of increasing TA muscle recovery rates over those achieved with empty nerve guides or CNG filled with cell-free 0.2% HA or 0.2% HAL. With regard to quantitative data for TA muscle function retrieved from nerves recovered under the respective conditions, we can conclude that repair with NNG + 0.7% M-HAL was equally supportive as repair with empty CNG, which, however, had overall a lower recovery rate.

As already mentioned above, PL muscle function cannot be considered fully developed at 120 days after surgery for M-HAL associated treatments and comparison to results from our previous studies would rather be a non-appropriate overestimation. It appears noteworthy, however, that adding 0.7% M-HAL was superior in supporting PL muscle recovery over 0.4% M-HAL in the current study. The percentile content of HA in our laminin-hyaluronic acid-based hydrogel luminal filler is influencing the viscosity or stiffness of the luminal content provided to invading cells and regenerating axons. From many studies, it can be concluded that optimally tuned nerve graft mechanical properties can give optimal support to repair Schwann cells as well as pro-regenerative macrophages as well as regenerating axons [9,26]. While the functional results presented here give already some indication of the quality of axonal outgrowth, and point towards more favorable properties of the higher concentrated 0.7% M-HAL, more details can be derived from our nerve morphometrical analyses.

Our histomorphometric analysis of semi-thin cross-sections obtained from nerve samples distal to the introduced grafts revealed that in none of the experimental groups, nerve fiber density (number of regenerated myelinated axons/mm^2^) recovered to a degree comparable to that seen in ANG-treated nerves (Figure 4). No significant differences in nerve fiber density could be shown for nerves reconstructed with different combinations of bioartificial grafts and M-HAL luminal filler in a concentration of 0.4% and 0.7% (Figure 4). However, CNG + 0.7% M-HAL- and NNG + 0.7% M-HAL-treated nerves demonstrated slightly higher distal nerve fiber densities compared to samples from the bioartificial nerve graft groups.

Interestingly, the animals demonstrating evocable CMAPs recorded from PL muscle, the more distal electrodiagnostic recording, from all experimental groups as well as the ANG-group showed higher values with regard to the nerve fiber density (Figure 2, triangles). This finding indicates that axonal outgrowth in number and distance, both contribute to successful functional recovery. It further underscores the interrelation between electrodiagnostic recordings and histomorphometric analysis and importance of comprehensive analysis in pre-clinical models [34].

With regard to a more detailed analysis of quantitative parameters of the regenerated myelinated axons, we and others before have analyzed the nerve fiber diameter distribution [11,35]. In the current study, the healthy reference samples showed a balanced nerve fiber diameter distribution, with a modal value of 8–9 µm. In comparison, regenerated axons in all experimental groups demonstrated a clear left shift towards smaller nerve fiber diameters (Appendix A). Myelinated axons regenerated through a 15 mm long ANG (gold standard control) presented a peak diameter at 4.5 µm. Axons regenerated over a distance of 15 mm through different combinations of bioartificial grafts and M-HAL luminal filler demonstrated a diameter peak at only 3–3.5 µm. Accordingly, the bioartificial grafts more clearly deviate from the fiber distribution of a healthy nerve. Also, in comparison to our previous results obtained from empty chitosan-based nerve graft repair and reconstruction of the 15 mm nerve gap with lower concentrated HA or HAL [30,31], we detected interesting differences (Figure 6). The reconstruction of the nerve defect with empty bioartificial nerve guides lead to a higher percentage of myelinated fibers with a diameter higher than 4 µm compared to grafts filled with 0.2% HA, 0.2% HAL, 0.4% M-HAL, and 0.7% M-HAL. In correlation to the functional motor recovery, this indicates that at least 15% to ≥20% of the myelinated fibers need to achieve a diameter >4 µm for supporting detectable motor recovery. Although we have to consider the overall recovery rates being lower for empty nerve guides and nerve guides filled with lower concentrated hydrogel than for conduits filled with higher concentrated M-HAL, this finding once again points towards the importance of tuning the hydrogel fillers towards the optimal mechanical properties for supporting axonal regeneration [9,26]. 

In view of the fact that optimized axonal regeneration crucially depends on the invasion of the bioartificial grafts by e.g., repair Schwann cells [18,19,26,30], perineurial fibroblasts [36], or pro-regenerative cells from the immune system [14,37,38], it is also important to consider the interaction of the provided bioartificial graft with regeneration supporting cells types. It was recently demonstrated that collagen-based nerve guides with an optimized internal 3D structure could be enriched with autologous Schwann cells for improving regeneration of an acutely repaired 13 mm sciatic nerve defect in the rat [39]. Especially the well-orchestrated interplay between the different cell types is crucial for an optimized support of axonal regeneration [37]. Schwann cells (SC) from the repair phenotype play the most crucial, centralized role in successful peripheral nerve regeneration [18,19,26]. After injury, Schwann cells secrete cytokines and interleukins, while myelin expression is inhibited for the time of early axonal outgrowth by the expression of the transcription factor Sox2 [40].

Luminal filler hydrogels may serve as a cell carrier system for introducing regeneration supporting cells directly together with the bioarticial nerve graft [26]. In this regard, we evaluated previously non-modified 0.2% HAL, but found it not valuable enough for supporting a repair phenotype of cultured Schwann cells in vitro [30]. In the current study, we cultured again purified neonatal Schwann cells (SCs) within 0.4% M-HAL for 3 days in vitro and then performed gene expression analysis for some selected genes. We could not perform the same in vitro test with 0.7% M-HAL because its rather high viscosity did not allow for gently suspending SCs within the gel and further resulted in difficulties for isolating total RNA (data not shown). Also when culturing SCs in 0.4% M-HAL, cell density was visibly reduced after the first day of seeding in comparison to the other culture conditions (regular medium, 0.2% HA, 0.2% HAL, 0.4% HA; Figure 7, *DIV 1*), but the cell density after three days in vitro (Figure 7, *DIV 3*) was not significantly different from that in the other conditions anymore. This indicates that as soon as the cells get arranged in M-HAL, they survive and proliferate. Furthermore, this finding demonstrates again that initially cell-free luminal filler hydrogels need to provide mechanical properties and in vivo degradation behavior that would allow timely invasion of regeneration promoting cells into the bioarticial grafts [26]. With regard to the results obtained from our current in vivo evaluation of 0.4% M-HAL and 0.7% M-HAL, we have to consider that properties were not optimal for primarily supporting cell invasion but rather supported adherence and guidance of axonal sprouts. This effect is likely attributable to the laminin peptide content of M-HAL [9,20,30]. Furthermore, our gene expression analysis demonstrated that SCs cultured in 0.4% M-HAL had a significantly reduced gene expression in comparison to control for the neurtrophic factor genes *gndf*, *fgf-2,* and *ngf* and also gene expression of *sox2* and *c-jun*, transcription factors typically elevated in repair Schwann cells [19,40] was reduced (Figure 8). On the contrary, the gene expression of the neurotrophic factor gene *bdnf* showed a trend to be slightly elevated in the 0.4% M-HAL condition than in the other three groups, although the difference was not statistically significant. We can only speculate that expression of brain derived neurotrophic factor, BDNF, has also been elevated within the M-HAL composite bioartificial nerve guides in vivo, and may have accounted for the better outcome in regeneration of myelinated functional motor axons discussed above. Indeed, BDNF plays a role as myelination factor in peripheral nerve regeneration [41].

At this point, it seems appropriate commenting on the fact 0.7% M-HAL did not provide optimal properties for our in vitro Schwann cell cultures, but was still evaluated and supportive of axonal regeneration in vivo. We have experienced before that another hydrogel perfectly supporting Schwann cell and even dorsal root ganglion neurite outgrowth in vitro was not at all supportive to axonal regeneration in vivo [29]. Therefore, we do not consider in vitro studies with M-HAL predictive enough for the in vivo approach, and did still perform the in vivo evaluation. In addition, because, as already mentioned above, indications from the literature exist that more complex structural support within nerve guidance conduits could increase functional outcome after nerve gap repair [26]. Our results indicate that although isolated neonatal rat Schwann cells in culture could not easily and gently be suspended in 0.7% M-HAL, it did not impair, but rather support axonal growth in vivo. This in consequence, indicates that Schwann cells were also able to migrate into the gel in vivo, because this is commonly known to precede axonal ingrowth into a nerve guide [42].

With regard to a recent report describing more pronounced beneficial effects of hyaluronic acid-laminin hydrogel in a delayed repair approach for 25 mm tibial nerve defects in the rabbit [28], we want to emphasize that the gel evaluated in the current study should be evaluated for its value for a delayed repair as well. We have demonstrated before that chitosan-based nerve guides provided a better substrate for delayed repair in the 15 mm sciatic nerve gap rat model than the procedure of grafting autologous muscle-in-vein grafts [31]. In acute repair in the same model, however, repair with bioartificial grafts was less effective than autologous repair approaches [31]. However, other reports also exist for muscle-in-chitosan tubular graft repair that demonstrated less positive effects in comparison to empty nerve guides in the delayed repair approach [35]. This, once again, underscores the importance of adjusting biomaterial properties appropriately for the nerve repair approach in focus. 

From the current study, we can, at first, conclude that the achievable benefit of adding 0.4% or 0.7% M-HAL as luminal filler, to otherwise empty nerve guides, did not significantly differ with regard to the material the bioartificial nerve graft was made of. Therefore, both collagen- or chitosan-based nerve guides appear to be equally suitable for a combination with M-HAL luminal filler hydrogel. Second, both concentrations of M-HAL provided a better support for axonal and functional recovery than 0.2% HAL in chitosan-based nerve guides before [30]. Finally, in future studies, for comprehensive pre-clinical testing, a comparison of acute and delayed repair approaches of critical gap length defects in the same model should be considered,. The latter is of importance, because the demands a nerve has on material properties of a bioartificial nerve graft may significantly differ between the acutely lesioned and the chronically lesioned nerve.

## 4. Materials and Methods 

### 4.1. 0.4% M-HAL and 0.7% M-HAL Preparation

M-HAL was prepared prior to use, based on 10 μg/mL laminin (synthesized at Bachem, Bubendorf, Switzerland), and 0.4% (*w*/*v*) or 0.7% (*w*/*v*) hyaluronic acid (Lifecore Biomedical, Chaska, MN, USA) [28]. HA stock solutions were prepared by dissolving 40 mg or 70 mg of high molecular weight HA (1.67 MDa) in 10 mL phosphate-buffered saline (PBS, Biochrom GmbH, Berlin, Germany) The 0.4% (*w*/*v*) and 0.7% (*w*/*v*) HA solutions were stored at 4 °C. The gel was stored on ice until use. 

### 4.2. Animals and Surgery Procedure

The animal experiments were conducted in accordance with the German animal protection regulations and European Communities Directive 2010/63/EU for the protection of animals for experimental purposes. The experiments were previously approved by the Animal Care Committee of Lower Saxony, Germany (approval code: 33.12 42502-04-16/2320; approval date: 30.11.2016) in accordance with the local Institutional Animal Care and Research Advisory.

In this study, we used 36 young adult female Lewis rats (LEW/OrlRj, mean body weight at the day of surgery: 195.7± 8.79 g), which were obtained from Janvier Labs SAS (Le Genest-Saint-Isle, France) at an age of 10 weeks. Group size was calculated by power analysis with Graphpad Statmate 2.0 software (GraphPad Software Inc., La Jolla, CA, USA), based on data from previous investigation of another hydrogel [29], with effect size >1, type I error: α: 0.05, *p* < 0.05, and type II error: β: 0.2, power 50%.

During the experiment, the animals were housed in groups of four in an enriched environment, under uniform housing conditions (22.2 °C; humidity 55.5%; light/dark cycle 14 h/10 h). Food and water were provided ad libitum. The drinking water was supplemented with amitriptyline hydrochloride (13.5 mg/kg/day, Amitriptylin-neuraxpharm^®^, Neuraxpharm Arzneimittel GmbH, Langenfeld, Germany) from two weeks ahead of the surgery until the study was completed, to prevent self-mutilation [43]. Health states of the animals were monitored every two to three days.

All surgeries were performed on 13-week old rats under deep anesthesia (intraperitoneal injection of chloral hydrate, 370 mg/kg, Sigma-Aldrich, St. Louis, MO, USA) and aseptic conditions. Throughout the surgery, the animals were placed on a heating pad to mitigate the decrease in body temperature. Additionally, a heating lamp was used during anesthesia induction and recovery phase. Rectal body temperature was measured before and after surgery in order to ensure a body temperature not lower than 36.5 °C. The recovery phase was supported by applying 2 mL of electrolyte solution (E148G1 Pad; Serumwerk Bernburg AG, Bernburg, Germany) subcutaneously. A sufficient level of analgesia was achieved by subcutaneous injection at a dosage of 0.5 mg/kg Butorphanol (Torbugesic^®^, Pfizer Inc., Brooklyn, NY, USA).

The surgical procedure was carried out in accordance with studies previously published [11,29,30]. Briefly, the left sciatic nerve was accessed at mid-thigh level. Bupivacaine (0.25%, Carbostesin^®^, AstraZeneca plc, Cambridge, UK) and lidocaine (2%, Xylocain^®^, AstraZeneca plc, Cambridge, UK) were topically applied on the exposed sciatic nerve 5 min before nerve transection, in order to ensure sufficient analgesia. Using a micro scissor, the first incision was applied 5 mm distal to the gluteus muscle’s aponeurosis. A 15 mm nerve defect was established for autologous nerve grafts (ANGs). The nerve graft was flipped over, rotated 180°, and sutured distal and proximal, each with three epineural 9-0 stitches spaced approximately 120° apart. A 13 mm nerve defect was established in all experimental groups and reconstructed with a 19 mm artificial nerve graft, generating a 2 mm overlap of both nerve ends (group description, Table 1).

Ahead of luminal filling, NeuraGen^®^ Nerve Guides (Integra LifeSciencesn North Billerica, MA, USA) and Reaxon^®^ Nerve Guides (Medovent GmbH, Mainz, Germany) were preconditioned in 0.9% sodium chloride rinsing solution (NaCl 0.9%, B. Braun Melsungen AG, Melsungen, Germany) for at least 20 min. Before tying the knot at the distal suture of the artificial nerve grafts, 60–120 µL of M-HAL luminal filler were applied. Wounds were closed with 3–4 resorbable single button sutures in the femoral biceps muscle (3-0 Polysorb, UL-215, Covidien, Dublin, Ireland) followed by 3–4 nonresorbable mattress sutures (4-0 EthilonTMII, EH7791H, Ethicon, Somerville, NJ, USA) for skin closure.

For ethical and animal welfare reasons (reduction of animal numbers used), we have not investigated again in the current study experimental groups we have already investigated before in the same model (15 mm rat sciatic nerve gap, animals from the same breeder, strain, sex and matched age). However, we used the previously obtained data for a more comprehensive evaluation of our new data derived from the current study. These previously obtained data have been published in Stößel et al. 2018 [31], and in Dietzmeyer et al. 2020 [30]. Thereby, repetitive evaluation of nerve reconstruction with empty chitosan-based nerve guides (CNG) as well as chitosan-based nerve guides filled with hyaluronic acid (HA) or with the original non-modified HAL both in a concentration of 0.2% has been avoided.

### 4.3. Functional Evaluation

The progress of functional motor recovery was monitored by serial non-invasive electrophysiological measurements 60, 90, and 120 days after nerve reconstruction. The muscle weight ratio (MWR) for lower hind limb muscles was determined after completion of the 120-day observation period. All surgery was performed by KHT and evaluators of regeneration progress were blinded to the conditions of sciatic nerve reconstruction applied.

Assessment of functional motor recovery with transcutaneous electrodiagnostic recordings: 

For the evaluation of ongoing muscle reinnervation 60, 90, and 120 days post-surgery, a portable electrodiagnostic device (Dantec^®^ Keypoint^®^ Focus device; Natus Europe GmbH, Planegg, Germany) was used. The electrodiagnostic managements were performed in accordance with previously published studies of our group [30,31]. Briefly, the animals were anesthetized and placed in a prone position on top of a thermostatic blanket. Only during recordings, the thermostatic blanket was turned off to avoid interfering signals. Monopolar needles (30 G, diameter 0.3 mm, length 10 mm; Alpine Biomed ApS, Tonsbakken, Denmark) were placed transcutaneous either at the sciatic notch, proximal to the injury site (proximal stimulation) or in the popliteal fossa (distal stimulation). The sciatic nerve was stimulated with single electrical pulses (100 µs, 1 Hz, 30% supramaximal level). The evocable compound muscle action potentials (CMAPs) of the tibialis anterior (TA) and plantar (PL) muscles were recorded by means of another pair of monopolar needles inserted transcutaneous into the muscle belly of the TA or PL, respectively. The reference electrode was inserted at the distal phalange of the fourth toe, the ground electrode at the knee. Reference values for each animal were obtained from the non-lesioned, right hind limb, if required.

Functional motor recovery was described qualitatively in terms of recovery rates (%) that were calculated from the number of animals per group displaying evocable CMAPs. Quantitative parameters were derived by analyzing the area under the curve (AUC) of the negative peaks of the CMAP *M*-waves and the amplitude of the recorded CMAP signals. CMAP amplitude ratio was calculated as a quantitative measure for the regeneration degree (CMAP amplitude values from the lesion side divided by the non-lesioned side. The CMAP AUC was used for calculating the percentage of functional axon loss between the non-lesioned healthy and the regenerated nerve by the formula: Axon loss (%) = AUC_healthy_ − AUC_regenerated_/AUC_healthy_ × 100. For the statistical analyses of electrodiagnostic measurements (CMAP amplitude area, axon loss, and CMAP amplitude ratio), all animals reaching the endpoint of the analysis were considered. Two animals died due to anesthesia complications after the first functional evaluation at 60 days post-surgery (ANG-group and CNG + 0.4% M-HAL-group).

Values for CMAP amplitude area and amplitude ratio were set to 0 for certain animals, which did not show evocable CMAPs, whereas the respective values for axon loss were set to 100%. 

Muscle weight ratio: After the last electrophysiological recording at 120 days post-surgery and harvest of nerve tissue for histology (see below), the anaesthetized animals were sacrificed in carbon dioxide atmosphere and by cervical dislocation after deep anesthesia was induced. The gastrocnemius (GC) and TA muscle of both hind limbs were excised and weighed. The MWR was calculated dividing the muscle weight of the lesioned side by the muscle weight of the non-lesioned side.

### 4.4. Nerve Histomorphometry 

Histomorphometrical analysis of the regenerated nerves was performed in accordance with a previously described protocol [11]. The samples, 5 mm in length, were dissected from the continuing nerve, distal to the inserted nerve graft. The healthy, contralateral sciatic nerve served as control (*n* = 3). The dissected nerve segments were transferred in Karnovsky fixative (2% PFA, 2.5% glutaraldehyde in 0.2 M sodium cacodylate buffer, pH 7.3) for 24 h. Afterwards, the samples were rinsed with 0.1 M sodium cacodylate buffer supplemented with 7.5% sucrose and post-fixed in 1% osmium tetroxide for 1.5 h. Staining of the myelin sheath was performed in 1% potassium dichromate (24 h), 25% ethanol (24 h), and hematoxylin (0.5% hematoxylin in 25% ethanol, for 24 h) [44]. Subsequently, the samples were embedded in Epon and 1 µm thick semi-thin cross-sections were cut. The sections were stained using toluidine blue to enhance the coloring of the myelin sheaths. The cross-sections were analyzed using a BX50 microscope (Olympus Europa SE & Co. KG, Hamburg, Germany), equipped with a prior controller (MBF Bioscience, Williston, VT, USA), and Stereo Investigator software, version 11.04 (MBF Bioscience, Williston, VT, USA). 

The stereological evaluation was performed according to our previously published work [30,31,45]. In brief, two randomly selected sections of each sample were included in the following analysis. The cross-sectional area was determined (in 20× magnification) to calculate the total number of myelinated fibers (in 100× magnification) and the respective fiber density with a two-dimensional dissector (optical fractionator; grid size: 150 µm × 150 µm; counting frame size: 30 µm × 30 µm; counting “fiber tips” as suggested by colleagues [46])”. Stereological analysis also considered axons that were located outside the perineurium of the fascicles but within the epineurium of the nerve.

Stereological analysis was only performed if the fiber density of the section was high enough to ensure a valid analysis with the described method. Consequently the number of analyzed samples did not for all groups match the number of evaluated animals: non-lesioned control: *n* = 3; ANG: *n* = 5; NNG: *n* = 4; NNG + 0.4% M-HAL: *n* = 6; NNG + 0.7% M-HAL: *n* = 6; CNG + 0.4% M-HAL: *n* = 4; CNG + 0.7% M-HAL: *n* = 5. One sample from the CNG + 0.7% M-HAL was further excluded from consecutive nerve morphometrical analysis, because it was not showing an appropriate number of regenerated myelinated axons. 

For each animal, six to eight view fields (in 100× magnification) of regions showing a high fiber density were acquired to analyze the morphometry of the regenerated nerves. ImageJ (version 1.48; National Institutes of Health, Bethesda, MD, USA) extended by a *g*-ratio plug-in [30,31,47] was used to evaluate axon diameter, fiber diameter, myelin thickness, and the corresponding *g-*ratio. Depending on the regeneration of the sample, a total number of at least thirty to a maximum of eighty axons per animal was evaluated. The data was collected under the assumption that the axons show a circular shape. Samples that showed no quantifiable axonal regeneration in the stereological analysis were excluded from morphometrical analysis. 

### 4.5. Schwann Cell Culture

Schwann cell cultures at passage 11 were taken from storage in liquid nitrogen and thawed as required. These cells were previously derived from primary neonatal Schwann cells (SCs) obtained from Wistar RjHan:WI rat pups and cultured and purified following a previously published protocol [48] and in accordance to [30]. In brief, sciatic nerves were collected and enzymatically digested. The separated cells were cultured in poly-l-lysine (PLL)-coated dishes containing Dulbecco’s modified Eagle’s medium supplemented with 0.1 µM Forskolin, 1% Pen/strep, 2 mM L-glutamine, 1 mM sodium pyruvate, and 10% fetal calf serum (FCS) (all from Thermo Fisher Scientific, Waltham, MA, USA) at 37 °C and 5% CO_2_ in a humidified atmosphere. PLL coating was performed by covering the culture dishes with PLL for 45 min at room temperature. After removing the PLL, plates were washed 2 times with Ampuwa^®^ (Fresenius Kabi, Bad Homburg vor Höhe, Germany). To prevent excessive fibroblast contamination, 1 mM of arabinoside-c (Sigma-Aldrich, St. Louis, MO, USA) was added at 2 and 3 days in vitro, respectively. The cells were finally purified to >90% by immunopanning, purity was controlled in immunocytochemistry, and cells propagated in medium as described above, just with an increased concentration of 2 µM Forskolin [30]. 

### 4.6. Quantitative RT-PCR

Neonatal rat SCs were cultured in a six-well plate (Nunc, Thermo Fisher Scientific, Waltham, MA, USA) containing the above described medium (see section Primary Schwann cell culture), serving as control. The initial cell density was 300,000 cells per well. Experimental conditions were cultured within 0.2% HA, 0.4% HA, or 0.4% M-HAL. We excluded 0.7% M-HAL culture conditions from this analysis, because Schwann cells could not be suspended gently enough in this concentration. Further, appropriate amounts of total RNA could not be extracted. 

Medium was used for preparing the needed concentrations. The respective solutions were prepared one day ahead, as described above (see section 0.4% M-HAL and 0.7% M-HAL preparation), except that the HA was dissolved in culture medium instead of PBS, vortexed thoroughly and stored overnight at 4 °C. The 0.4% M-HAL was prepared freshly. 

Total RNA of 3 wells was analyzed after 3 days in vitro as described previously [30]. In brief, the cell culture supernatant was removed and the cells were treated according to the manufacturer’s guidelines (RNeasy Plus Mini Kit, Qiagen, Hilden, Germany). Equal amounts of mRNA were used for reverse transcription applying iScript^TM^ cDNA Synthesis Kit (Bio-Rad Laboratories, Inc., Hercules, CA, USA). The qRT-PCR was performed according to a previously describes protocol [49]. The reaction mixture was prepared in a MicroAmp reaction plate (Applied Biosystems) containing 5 µL diluted cDNA, 2 µL forward and reverse primer (1.75 µM each) and 7 µL Power SYBRgreen PCR Master Mix (Applied Biosystems, Waltham, MA, USA). The qPCR was performed on a StepOnePlus thermocycler (Applied Biosystems, Waltham, MA, USA): PCR was conducted for 40 cycles (15 s 95 °C and 1 min 60 °C) after an initial 10 min step of 95 °C. C_T_-values were calculated with StepOne-software version 2.3 using a constant cycle threshold of 0.2. Calculation of relative amount of transcript in cDNA levels was performed and normalized to the housekeeping gene peptidylprolyl isomerase A (*ppia*). The primer sequences are given in Table 4. We performed this experiment in biological triplicates and technical duplicates.

### 4.7. Statistical Analysis

Statistical analysis of data obtained from the current study was performed using Excel Version 2010 (Microsoft Corporation, Redmond, WA, USA) and GraphPad Prism version 8.01 (GraphPad Software Inc., USA). Normal distribution of all datasets was examined with Shapiro-Wilk test. If normal distribution was given, one-way study of variation (ANOVA) with Tukey’s multiple comparisons post-hoc test was used to identify significant differences. If normal distribution was rejected, the data was subjected to Kruskal–Wallis test with Dunn’s multiple comparisons post-hoc test was applied. A two-way ANOVA with Tukey’s multiple comparisons post-hoc test was used to compare the electrodiagnostic recordings during the observation period. All results are presented as percentages or mean ± SEM, as indicated in the respective tables or figures.

Statistical significance was defined and indicated as follows: one symbol = *p* < 0.05; two symbols = *p* < 0.01; three symbols = *p* < 0.001. 

## Figures and Tables

**Figure 1 ijms-22-06554-f001:**
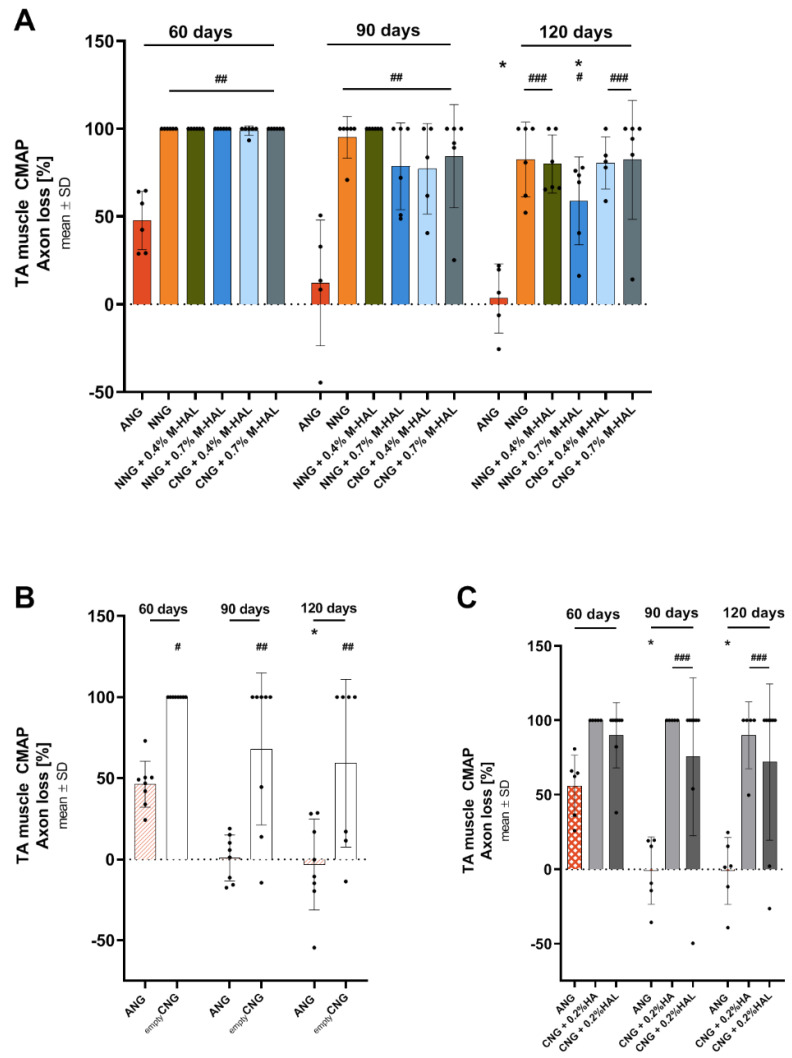
Graphical presentation of results from motor functional recovery evaluation of the tibialis anterior (TA) muscle. Bar graphs depict the percentile axons loss as mean ± SD and dots show values of each animal analyzed. The axon loss was calculated from evocable compound muscle action potential (CMAP) area under the curve. (**A**) Axon loss [%] as derived from the analysis in the current study; (**B**) axon loss [%] as derived from the analysis in Stößel et al. 2018 [31]; (**C**) axon loss [%] as derived from the analysis in Dietzmeyer et al. 2020 [30]. Significant differences are indicated as follows: “*” indicates *p* < 0.05, vs. 60 days values within the same group; “#” indicates *p* < 0.05, “##” indicates *p* < 0.01, “###” indicates *p* < 0.001 vs. ANG group at the same time point.

**Figure 2 ijms-22-06554-f002:**
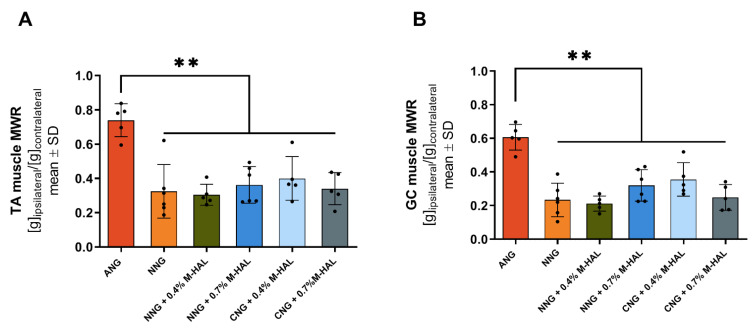
Graphical presentation of the results from lower limb muscle weight ratio (MWR) calculation at 120 days after reconstruction of the sciatic nerve. The MWRs were derived from comparing the weight of the TA muscle (**A**) and gastrocnemius (GC) muscle (**B**) collected from the non-lesioned to those collected from treated hind limb of each animal. Data was subjected to one-way ANOVA followed by Tukey’s multiple comparisons post-hoc test. ANG: *n* = 5; NNG: *n* = 6; NNG + 0.4% M-HAL: *n* = 5; NNG + 0.7% M-HAL: *n* = 6; CNG + 0.4% M-HAL: *n* = 5; CNG + 0.7% M-HAL: *n* = 5. Significant differences are indicated as ** *p* < 0.001 vs. ANG group.

**Figure 3 ijms-22-06554-f003:**
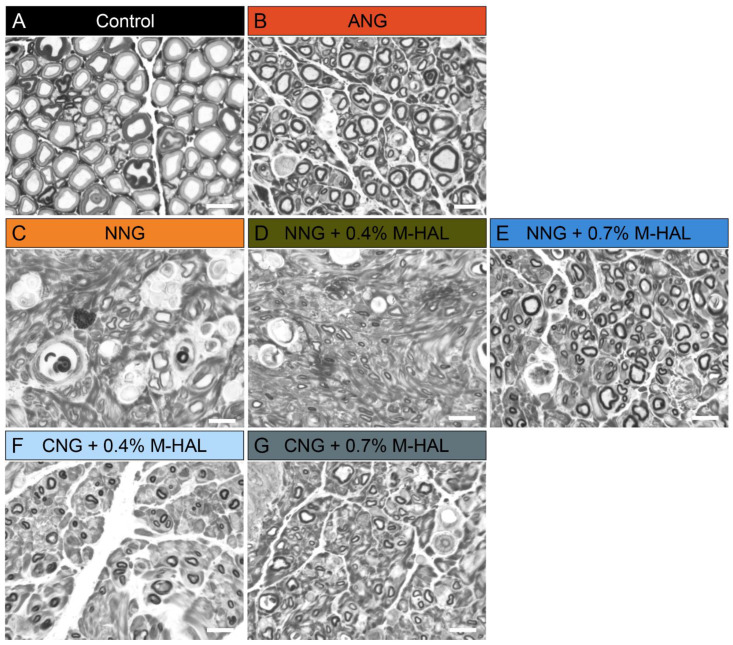
Representative photomicrographs of semi-thin, toluidine blue stained cross-sections of the distal nerve segments at 120 days post-surgery. (**A**) Non-lesioned control, (**B**) ANG, (**C**) NNG, (**D**) NNG + 0.4% M-HAL, (**E**) NNG + 0.7% M-HAL, (**F**) CNG + 0.4% M-HAL, (**G**) CNG + 0.7% M-HAL. Scale bars: 10 µm.

**Figure 4 ijms-22-06554-f004:**
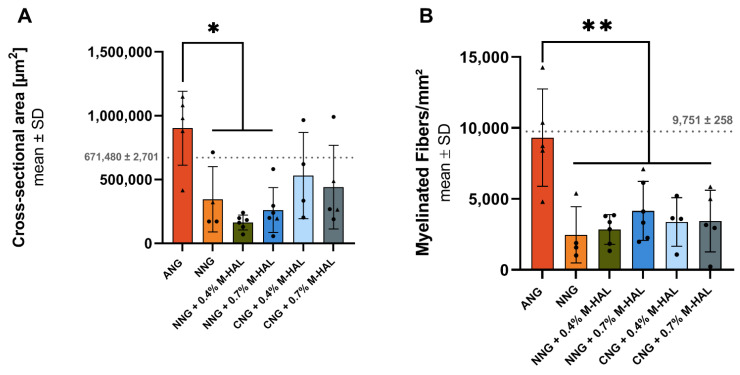
Graphical presentation of the results derived from stereological analysis of distal nerve segments at 120 days after reconstruction of the sciatic nerve. Bar graphs depict the mean ± SEM and dots or triangles show values of each animal analyzed. (**A**) Cross-sectional area of the analyzed nerve segment. (**B**) Nerve fiber density (myelinated fibers/mm^2^). Dotted lines indicate mean ± SD obtained from *n* = 3 non-lesioned distal rat sciatic nerve segments. (**A**,**B**) Data was subjected to parametric one-way ANOVA followed by Tukey’s multiple comparisons post-hoc test. “*” indicates *p* < 0.05. “**” indicates *p* < 0.01 compared to ANG group. Triangles represent animals showing recovery of evocable compound muscle action potentials in the plantar muscle. ANG: *n* = 5; NNG: *n* = 4; NNG + 0.4% M-HAL: *n* = 6; NNG + 0.7% M-HAL: *n* = 6; CNG + 0.4% M-HAL: *n* = 4; CNG + 0.7% M-HAL: *n* = 4.

**Figure 5 ijms-22-06554-f005:**
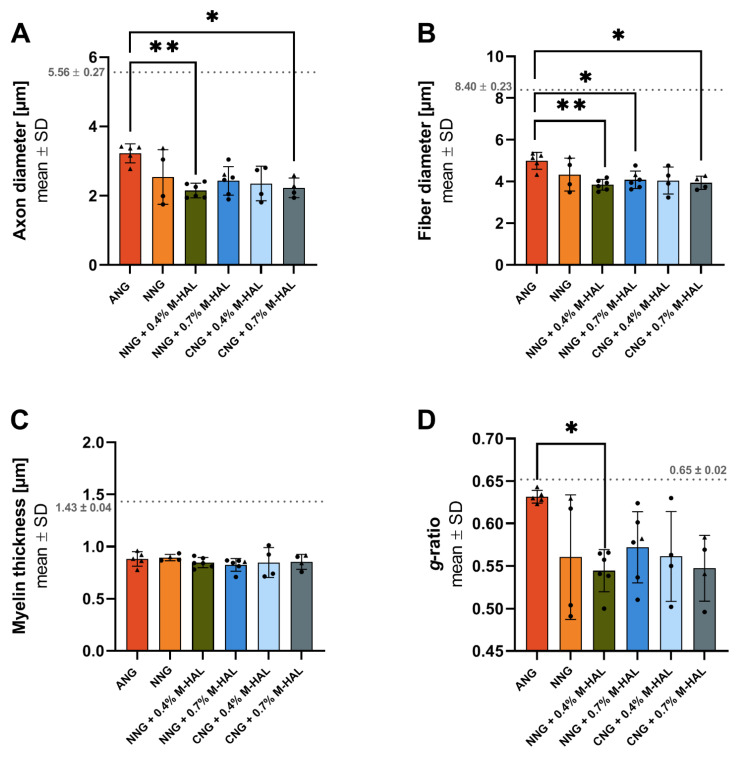
Graphical presentation of the results derived from morphometrical analysis of the distal sciatic nerve at 120 days post-surgery. Bar graphs depict the mean ± SD and dots or triangles show values of each animal analyzed. (**A**) Axon diameter, (**B**) Fiber diameter, (**C**) Myelin thickness, (**D**) *g*-ratio. Dotted lines indicated mean ± SD obtained from *n* = 3 non-lesioned distal rat sciatic nerve segments. (**A**,**B**,**D**) Data was subjected to parametric one-way ANOVA followed by Tukey’s multiple comparisons post-hoc test. (**C**) Data was subjected to nonparametric Kruskal-Wallis test followed by Dunn’s multiple comparisons post-hoc test. “*” indicates *p* < 0.05, “**” indicates *p* < 0.01 compared to ANG group. Triangles represent animals showing recovery of evocable compound muscle action potentials in the plantar muscle. ANG: *n* = 5; NNG: *n* = 4; NNG + 0.4% M-HAL: *n* = 6; NNG + 0.7% M-HAL: *n* = 6; CNG + 0.4% M-HAL: *n* = 4; CNG + 0.7% M-HAL: *n* = 4.

**Figure 6 ijms-22-06554-f006:**
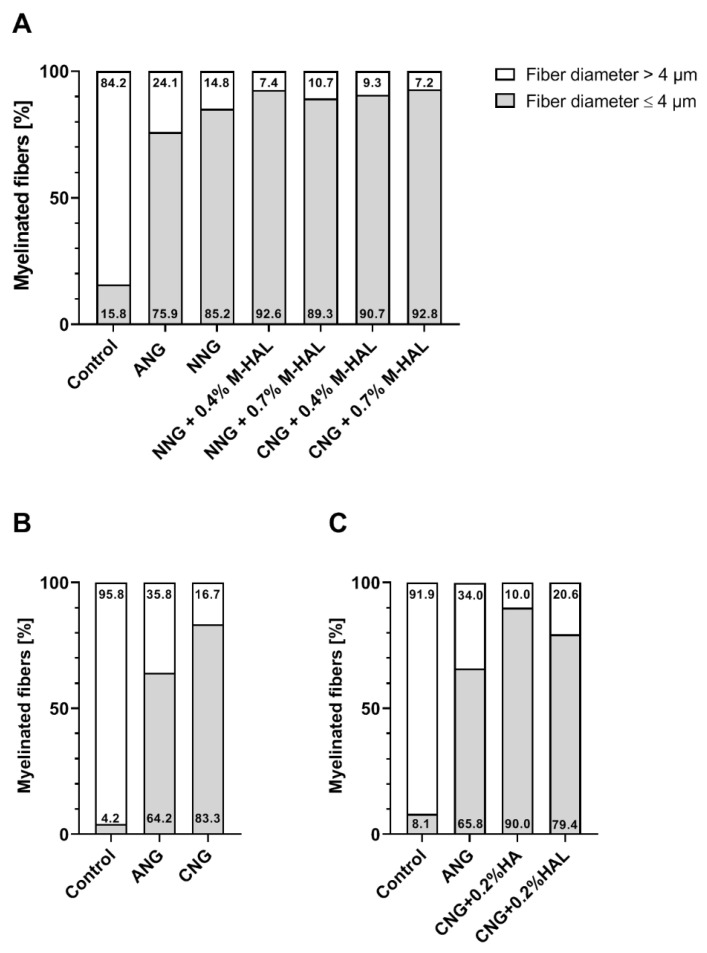
Bar graphs depicting the percentile distribution of nerve fiber diameters within the analyzed distal nerve segments at 120 days after reconstruction surgery. Stacked column graphs show the percentage of nerve fibers with a diameter ≥ 4µm (white) and ≤4 µm (light grey). The presented data originate from the analysis of the current study (**A**), the re-analysis of data published in Stößel et al. 2018 [31] (**B**), and the re-analysis of data published in Dietzmeyer et al. 2020 [30] (**C**). The number of evaluated animals was: (**A**) Control: *n* = 3; ANG: *n* = 5; NNG: *n* = 4; NNG + 0.4% M-HAL: *n* = 6; NNG + 0.7% M-HAL: *n* = 6; CNG + 0.4% M-HAL: *n* = 4; CNG + 0.7% M-HAL: *n* = 4, (**B**) Control: *n* = 3; ANG: *n* = 3; CNG: *n* = 3, and (**C**) Control: *n* = 6; ANG: *n* = 6; CNG + 0.2% HA: *n* = 1; CNG + 0.2% HAL: *n* = 2.

**Figure 7 ijms-22-06554-f007:**
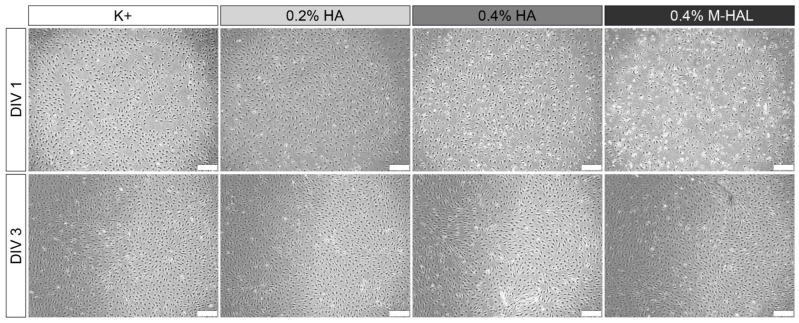
Representative phase-contrast photomicrographs of rat primary neonatal Schwann cells (SC) seeded in culture medium (K+), 0.2% hyaluronic acid (0.2% HA), 0.4% HA, and 0.4% modified hyaluronic acid-laminin-hydrogel (0.4% M-HAL). Initial cell density was 300,000 SCs per well. While the cell density after 1 day (DIV 1, top row) seemed to be lower in 0.4% M-HAL, a similar dense cell layer was seen after three days in vitro (DIV 3, bottom) for all conditions. Scale bar: 200 µm.

**Figure 8 ijms-22-06554-f008:**
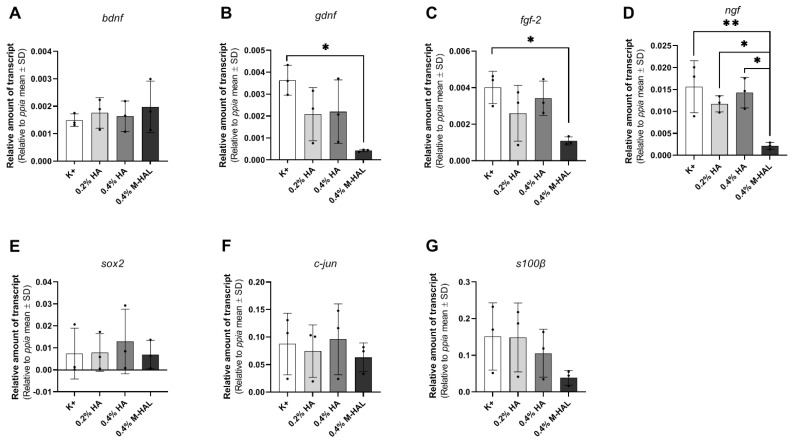
Graphical presentation of gene expression analysis in neonatal rat Schwann cells (SCs) after three days in vitro under different culture conditions. Bar graphs display relative amounts of transcripts to the housekeeping gene *ppia* as mean ± SD. Dots represent the results of *n* = 3 replicates. SCs were cultured in medium (K+), 0.2% hyaluronic acid (HA), 0.4% hyaluronic acid (HA) or 0.4% modified hyaluronic acid-laminin-hydrogel (M-HAL) and gene expression analyzed for: (**A**) *bdnf*, (**B**) *gdnf*, (**C**) *fgf-2*, (**D**) *ngf*, (**E**) *sox2*, (**F**) *c-jun*, and (**G**) *s100ß*. Parametric one-way ANOVA followed by Tukey’s multiple comparisons post-hoc test was applied to detect significant differences * *p* < 0.05 and ** *p* < 0.01 vs. 0.4% M-HAL.

**Table 1 ijms-22-06554-t001:** In vivo experimental design.

Group	Nerve Graft	Luminal Filler	Number of Animals
ANG	Autologous nerve graft		6
NNG	NeuraGen^®^ Nerve Guide		6
NNG + 0.4% M-HAL	NeuraGen^®^ Nerve Guide	0.4% M-HAL	6
NNG + 0.7% M-HAL	NeuraGen^®^ Nerve Guide	0.7% M-HAL	6
CNG + 0.4% M-HAL	Reaxon^®^ Nerve Guide	0.4% M-HAL	6
CNG + 0.7% M-HAL	Reaxon^®^ Nerve Guide	0.7% M-HAL	6

**Table 2 ijms-22-06554-t002:** Evocable Compound Muscle Action Potential (CMAP) recorded from the anterior tibial (TA) muscle displaying functional motor recovery. Recovery rate indicates number and percentage of animals per group showing evocable CMAP. Further, CMAP amplitude areas, functional axon loss, and CMAP amplitude ratio, were calculated from noninvasive electrodiagnostic recordings at 60, 90, and 120 days after reconstruction surgery. Data were subjected to two-way ANOVA followed by Tukey’s multiple comparisons post-hoc test. “*” indicates *p* < 0.05 vs. 60 days values within the same group; “#” indicates *p* < 0.05 vs. ANG values at the same time point.

	Group	Recovery Rate Animals (Percentage)	CMAP Amplitude Area [ms * mV] Mean ± SD	Axon Loss [%] Mean ± SD	CMAP Amplitude Ratio Mean ± SD
**60 days** **post-surgery**	ANG	6/6 (100%)	20.07 ± 6.382	47.75 ± 16.62	0.37 ± 0.10
	NNG	0/6 (0%)	0.00 ± 0.00 #	100.00 ± 0.00 #	0.00 ± 0.00 #
	NNG + 0.4% M-HAL	0/6 (0%)	0.00 ± 0.00 #	100.00 ± 0.00 #	0.00 ± 0.00 #
	NNG + 0.7% M-HAL	0/6 (0%)	0.00 ± 0.00 #	100.00 ± 0.00 #	0.00 ± 0.00 #
	CNG + 0.4% M-HAL	1/6 (16.7%)	0.43 ± 1.04 #	98.90 ± 2.71 #	0.01 ± 0.02 #
	CNG + 0.7% M-HAL	0/6 (0%)	0.00 ± 0.00 #	100.00 ± 0.00 #	0.00 ± 0.00 #
**90 days** **post-surgery**	ANG	5/5 (100%)	26.21 ± 10.71	12.16 ± 35.88	0.61 ± 0.16
	NNG	1/6 (16.7)	1.45 ± 3.55 #	95.14 ± 11.90 #	0.03 ± 0.08 #
	NNG + 0.4% M-HAL	0/6 (0%)	0.00 ± 0.00 #	100.00 ± 0.00 #	0.00 ± 0.00 #
	NNG + 0.7% M-HAL	3/6 (50%)	6.38 ± 7.40 #	78.62 ± 24.80 #	0.12 ± 0.14 #
	CNG + 0.4% M-HAL	3/5 (60%)	6.80 ± 7.69 #	77.21 ± 25.78 #	0.13 ± 0.13 #
	CNG + 0.7% M-HAL	3/6 (50%)	4.67 ± 8.77 #	84.35 ± 29.38 #	0.08 ± 0.15 #
**120 days** **post-surgery**	ANG	5/5 (100%)	30.40 ± 6.20	3.29 ± 19.70 *	0.83 ± 0.24 *
	NNG	3/6 (50%)	5.51 ± 6.70 #	82.48 ± 21.30 #	0.10 ± 0.13 #
	NNG + 0.4% M-HAL	4/6 (66.7%)	6.31 ± 5.23 #	79.93 ± 16.63 #	0.12 ± 0.10 #
	NNG + 0.7% M-HAL	6/6 (100%)	12.91 ± 7.88 *, #	58.95 ± 25.06 *, #	0.25 ± 0.14 *, #
	CNG + 0.4% M-HAL	4/5 (80%)	6.12 ± 4.66 #	80.53 ± 14.83 #	0.15 ± 0.10 #
	CNG + 0.7% M-HAL	3/6 (50%)	5.58 ± 10.65 #	82.27 ± 33.89 #	0.13 ± 0.22 #

**Table 3 ijms-22-06554-t003:** Evocable Compound Muscle Action Potential (CMAP) recorded from the plantar (PL) muscle displaying functional motor recovery. Recovery rate indicates number and percentage of animals per group showing evocable CMAP. Further, CMAP amplitude areas, functional axon loss, and CMAP amplitude ratio, were calculated from noninvasive electrodiagnostic recordings at 60, 90, and 120 days after reconstruction surgery. Data were subjected to two-way ANOVA followed by Tukey’s multiple comparisons post-hoc test. “*” indicates *p* < 0.001 vs. 60 days values within the same group; “&” indicates *p* < 0.001 vs. 90 days values within the same group; “#” indicates *p* < 0.001 vs. ANG values at the same time point.

	Group	Recovery Rate Animals (Percentage)	CMAP Amplitude Area [ms * mV] Mean ± SD	Axon Loss [%] Mean ± SD	CMAP Amplitude Ratio Mean ± SD
**60 days** **post-surgery**	ANG	4/6 (66.7%)	0.12 ± 0.15	94.02 ± 7.80	0.04 ± 0.05
	NNG	0/6 (0%)	0.00 ± 0.00	100.00 ± 0.00	0.00 ± 0.00
	NNG + 0.4% M-HAL	0/6 (0%)	0.00 ± 0.00	100.00 ± 0.00	0.00 ± 0.00
	NNG + 0.7% M-HAL	0/6 (0%)	0.00 ± 0.00	100.00 ± 0.00	0.00 ± 0.00
	CNG + 0.4% M-HAL	0/6 (0%)	0.00 ± 0.00	100.00 ± 0.00	0.00 ± 0.00
	CNG + 0.7% M-HAL	0/6 (0%)	0.00 ± 0.00	100.00 ± 0.00	0.00 ± 0.00
**90 days** **post-surgery**	ANG	5/5 (100%)	1.23 ± 0.67 *	57.14 ± 23.32 *	0.25 ± 0.11 *
	NNG	0/6 (0%)	0.00 ± 0.00 #	100.00 ± 0.00 #	0.00 ± 0.00 #
	NNG + 0.4% M-HAL	0/6 (0%)	0.00 ± 0.00 #	100.00 ± 0.00 #	0.00 ± 0.00 #
	NNG + 0.7% M-HAL	0/6 (0%)	0.00 ± 0.00 #	100.00 ± 0.00 #	0.00 ± 0.00 #
	CNG + 0.4% M-HAL	0/5 (0%)	0.00 ± 0.00 #	100.00 ± 0.00 #	0.00 ± 0.00 #
	CNG + 0.7% M-HAL	0/6 (0%)	0.00 ± 0.00 #	100.00 ± 0.00 #	0.00 ± 0.00 #
**120 days** **post-surgery**	ANG	5/5 (100%)	1.19 ± 0.31 *	43.38 ± 14.75 *	0.40 ± 0.08 *, &
	NNG	1/6 (16.7%)	0.01 ± 0.02 #	99.56 ± 1.08 #	0.002 ± 0.004 #
	NNG + 0.4% M-HAL	0/6 (0%)	0.00 ± 0.00 #	100.00 ± 0.00 #	0.00 ± 0.00 #
	NNG + 0.7% M-HAL	1/6 (16.7%)	0.02 ± 0.05 #	98.95 ± 2.57 #	0.01 ± 0.01 #
	CNG + 0.4% M-HAL	0/5 (0%)	0.00 ± 0.00 #	100.00 ± 0.00 #	0.00 ± 0.00 #
	CNG + 0.7% M-HAL	2/6 (33.3%)	0.10 ± 0.23 #	95.24 ± 10.69 #	0.02 ± 0.05 #

**Table 4 ijms-22-06554-t004:** Primer sequences for RT-qPCR.

Gene	Forward Primer (5′ to 3′)	Reverse Primer (5′ to 3′)	Expected Amplicon Size (bp)
*ppia*	TGTGCCAGGGTGGTGACTT	TCAAATTTCTCTCCGTAGATGGACTT	69
*bdnf*	GGACATATCCATGACCAGAAAGAA	GCAACAAACCACAACATTATCGAG	89
*gdnf*	CCAGAGAATTCCAGAGGGAAAGGT	TCAGTTCCTCCTTGGTTTCGTAGC	124
*ngf*	ACCTCTTCGGACACTCTGGA	GTCCGTGGCTGTGGTCTTAT	168
*fgf-2*	GAACCGGTACCTGGCTATGA	CCAGGCGTTCAAAGAAGAAA	86
*c-jun*	GCCTGCCTCTCTCAACTATGT	TAGGACACCCAAACAAACAAAC	126
*sox2*	GTAAGAAAAATCTGAATGCTCA	CCTCCAGATCTCTCATAAAAGT	147
*s100b*	AGTACGTATTTGACATCAACAGT	GAGACGTCTACTGAGCAGAAT	113

## Data Availability

The datasets analyzed during this study are available from the corresponding author on request. Raw data are stored in the authors’ institutional repositories and will be accordingly provided upon request.

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
