# Peer review of "Modified Hyaluronic Acid-Laminin-Hydrogel as Luminal Filler for Clinically Approved Hollow Nerve Guides in a Rat Critical Defect Size Model"

_ijms, 2021, doi:10.3390/ijms22126554_

Round 1
Reviewer 1 Report
Present work discuss about use of modified Hyaluronic acid-laminin hydrogel for nerve repair in rat model. I think work is interesting and can be accepted with few modifications
1) Introduction doesn't provide any information about role of laminin or hyaluronic acid for its possibility of cellular regenerations or repairs. It says that they have done previous studies on 0.2% hyaluronic acid-laminin-hydrogel (HAL) as luminal filler of chitosan-based nerve guides in the acute repair of 15 mm critical length sciatic nerve defects in the adult rat.
But as a part of this paper, I request authors to kindly refer some interesting works and provide some small introduction to use of laminins and hyaluronic acids, ref: https://www.mdpi.com/1996-1944/10/8/929/htm
2) As previous study provided data on 0.2% hyaluronic acid-laminin-hydrogel (HAL). For this study authors have increased their concentration to 0.4% and 0.7% gels kindly provide some rational behind this in the introduction section along with clinical importance.
3) too long title, please simplify it
4) Although I understand that main agenda for the work was to determine the pharmacological response of the hydrogel. And I think work is really good. But it is important to provide rheological (viscosity) data for their prepared gel.
5) Please provide graphical abstract
Author Response
Authors’ point-by-point response to Reviewer #1:
Comment reviewer #1: Present work discuss about use of modified Hyaluronic acid-laminin hydrogel for nerve repair in rat model. I think work is interesting and can be accepted with few modifications.
Author response: We are very grateful to the reviewer for this positive comment on our manuscript and provide a point-by-point response below.
Comment reviewer #1: 1) Introduction doesn't provide any information about role of laminin or hyaluronic acid for its possibility of cellular regenerations or repairs. It says that they have done previous studies on 0.2% hyaluronic acid-laminin-hydrogel (HAL) as luminal filler of chitosan-based nerve guides in the acute repair of 15 mm critical length sciatic nerve defects in the adult rat.
But as a part of this paper, I request authors to kindly refer some interesting works and provide some small introduction to use of laminins and hyaluronic acids, ref: https://www.mdpi.com/1996-1944/10/8/929/htm
Author response: We thank the reviewer for this comment and extended the introduction by some sentences and literature references regarding the role of laminins and hyaluronic acid in peripheral nerve regeneration. Please see: line 65-76
“A major class of ECM components are laminins, which have been shown to support differentiation and directed migration of Schwann cells, axon myelination during the processes of peripheral nerve regeneration [21-23, DOI: 10.1083/jcb.97.3.772, DOI: 10.1083/jcb.200307068, DOI: 10.1016/j.biomaterials.2019.119335]. Hyaluronic acid (HA), another key component of the ECM, plays an important role in neural proliferation, differentiation, migration, survival, and cell signaling [24, DOI: 10.1155/2015/368584.368584]. As Schwann cells represent the glia cells of the peripheral nervous system and are as such crucially involved in maintenance and regeneration processes [19, DOI:10.3389/fncel.2019.00033], both, laminin and HA have been studied and demonstrated their potential as nanomedicals for supporting cellular regeneration [25, DOI: 10.3390/ma10080929]. How extracellular matrix components can work as instructive engineered microenvironments for peripheral nerve repair approaches has recently been reviewed in [26, DOI: 10.3389/fbioe.2021.674473]. Here the authors also highlight the good potential for clinical use of these molecules due to their well standardized and reproducible synthesis [26, DOI: 10.3389/fbioe.2021.674473 ].”
Comment reviewer #1: 2) As previous study provided data on 0.2% hyaluronic acid-laminin-hydrogel (HAL). For this study authors have increased their concentration to 0.4% and 0.7% gels kindly provide some rational behind this in the introduction section along with clinical importance.
Author response: We thank the reviewer for their constructive comment. We have added the respective information about clinical importance along with the modifications explained already above. Please see: lines 75-76.
“ … Here the authors also highlight the good potential for clinical use of these molecules due to their well standardized and reproducible synthesis [26, DOI: 10.3389/fbioe.2021.674473 ].”
Regarding the rational for increasing the concentration in M-HAL we added some more information to the fore-fore-last paragraph of the introduction as well. Please see: lines 83-88, it now reads:
“… Interestingly, a further modified formulation of the gel with increased concentration introduced into collagen-based nerve guides did well support axonal regeneration in a rabbit model of delayed repair of 25 mm tibial nerve defects [28]. This finding contributes to the view that complex structural support within nerve guidance conduits is a valuable condition for increasing functional outcome after nerve gap repair [26].
Comment reviewer #1: 3) too long title, please simplify it
Author response: We thank the reviewer for their advice. The manuscript is now entitled: “Modified hyaluronic acid-laminin-hydrogel as luminal filler for clinically approved nerve guides in a rat critical defect size model”.
Comment reviewer #1: 4) Although I understand that main agenda for the work was to determine the pharmacological response of the hydrogel. And I think work is really good. But it is important to provide rheological (viscosity) data for their prepared gel.
Author response: We appreciate the reviewer’s comment about the overall quality of our study design and manuscript. We also understand the request for rheological data (viscosity) of the gel from a more material scientist approach. Anyway, we hope the reviewer can agree with us on keeping the focus of our work. And further ask to except our apologies for not being able to provide the requested data, considering the short revision due date of 7 days only.
Comment reviewer #1: 5) Please provide graphical abstract
Author response: We thank the reviewer for motivating us in providing a graphical abstract, which we submit along with this revision.
Reviewer 2 Report
Dear authors, your manuscript is well written and complete, I really appreciated it. Moreover the experimental design is clear and Table 1 is very useful.
For the result section I have some requests or I require some clarifications:
- why did you performed only electrophysiology as motor functional evaluation? There were other non invasive tests that could have been used.
- In Figure 1, 6, 8 and S1 and S2, some writings are too small and need to be enlarged, for example in Figure 1 you have to enlarged:
- "60, 90 and 120 days" and the experimental groups.
- Legend figure 2, please add "at 120 days after reconstruction of the sciatic nerve" at the end of this sentence: "Graphical presentation of the results from lower limb muscle weight ratio (MWR) calculation."
- 2.3 Evaluation of nerve histomorphometry, lines 277-285 information to add to "material and methods section".
- 2.4. In vitro analysis of neurotrophic factor expression. The reason why you didn't use 0.7 % M-HAL should be also described in material and methods or result section and not only in the discussion. Nevertheless, it does not make sense that 0.7% M-HAL was used in vivo and not in vitro. If you have viscosity problem in vitro, why it should be used in vivo and why it is working? It should be better explained in the discussion.
"Materials and Methods"
- 4.2. Animals and surgery procedure: "In this study, we used 36 young adult female Lewis rats" how the number of animals was calculated? Please report sample size calculation.
- 4.6 Quantitative RT-PCR: "Total RNA of 3 wells was analyzed after 3 days in vitro as described previously" (line 805), do you performed this experiments in technical and biological triplicate? Since it seems that the experiment was performed only once in technical triplicate. If it is not, please repeat the in vitro experiments.
- Please add to Table 4 "Amplicon size (bp)" and "Amplification efficiency (%)"
- 4.7. Statistical Analysis: please replace SEM with standard deviation in all your data Figures.
Author Response
Authors’ point-by-point response to Reviewer #2:
Comment reviewer #2: Dear authors, your manuscript is well written and complete, I really appreciated it. Moreover the experimental design is clear and Table 1 is very useful.
Author response: We are very grateful to the reviewer for the kind appreciation of our work and provide a point-by-point response below.
For the result section I have some requests or I require some clarifications:
Comment reviewer #2: why did you performed only electrophysiology as motor functional evaluation? There were other non invasive tests that could have been used.
Author response: We thank the reviewer for this question. The reviewer is correct, beside the electrophysiology there are other noninvasive tests for functional motor evaluation (e.g. walking track, sciatic functional index (SFI), Sciatic static index (SSI)). And we have also used functional index evaluations in many of our previous studies. Form our experiences, and underscored by references given from other experts in the field (e.g., Navarro, X. Functional evaluation of peripheral nerve regeneration and target reinnervation in animal models: a critical overview. Eur J Neurosci 2015, 43, 271-281, doi:10.1111/ejn.13033), we do not judge this a valuable technique when it comes to long gap repair. The indices are easily to be biased by the development of hamstring contractures and will not result in objective and reliable data like obtainable by electrodiagnostic measurements. Please kindly referr to DOI: 10.1002/mus.22023 and DOI: 10.1016/j.biomaterials.2013.08.074.
Comment reviewer #2: In Figure 1, 6, 8 and S1 and S2, some writings are too small and need to be enlarged, for example in Figure 1 you have to enlarged: "60, 90 and 120 days" and the experimental groups.
Author response: Thank you for your thoroughly reviewing our manuscript, we are happy to submit revised Figure 1, 6, and 8 in the revised manuscript and Figure S1 and S2 in the revised supplementary materials file.
Comment reviewer #2: Legend figure 2, please add "at 120 days after reconstruction of the sciatic nerve" at the end of this sentence: "Graphical presentation of the results from lower limb muscle weight ratio (MWR) calculation."
Author response: We apologize for not being clear here, during revision, we have added the missing information to the legend of figure 2. Please see: lines 271-272.
Comment reviewer #2: 2.3 Evaluation of nerve histomorphometry, lines 277-285 information to add to "material and methods section".
Author response: We thank the reviewer for this comment. Since we found the information also to be helpful for the reader in the results again we now present it twice, although shortened in the results section. Please see: lines 283-290 and lines 786-792.
Comment reviewer #2: 2.4. In vitro analysis of neurotrophic factor expression. The reason why you didn't use 0.7 % M-HAL should be also described in material and methods or result section and not only in the discussion. Nevertheless, it does not make sense that 0.7% M-HAL was used in vivo and not in vitro. If you have viscosity problem in vitro, why it should be used in vivo and why it is working? It should be better explained in the discussion.
Author response: We well understand the reviewers concern and have added some information to materials and methods for increasing clarity in this point. Please see: lines 825-828 of the revised manuscript.
With regard to the “viscosity problem”, we have extended a bit the discussion. We have been experiencing before that hydrogel perfectly supporting Schwann cell and dorsal root ganglion neurite outgrowth in vitro, was not at all supportive to axonal regeneration in vivo (DOI: 10.3727/096368915X688010). Therefore, we do not find in vitro studies with hydrogels enough predictive for the in vivo approach, and did still perform the in vivo evaluation. Also because indications from the literature exist, that complex structural support within nerve guidance conduits can be a valuable condition for increasing functional outcome after nerve gap repair (DOI: 10.3389/fbioe.2021.674473). Please see the revised text in lines 582-583, and 607-620.
"Materials and Methods"
Comment reviewer #2: 4.2. Animals and surgery procedure: "In this study, we used 36 young adult female Lewis rats" how the number of animals was calculated? Please report sample size calculation.
Author response: We thank the reviewer for this question and are happy to provide the requested information with the revision.
“Group size was calculated by power analysis with Graphpad Statmate 2.0 software (GraphPad Software Inc., USA), based on data from previous investigation of another hydrogel (DOI: 10.3727/096368915X688010), with effect size >1, type I error: α: 0.05, p < 0.05, and type II error: β: 0.2, power 50%.” Please see lines 662-665 of the revised manuscript.
Comment reviewer #2: 4.6 Quantitative RT-PCR: "Total RNA of 3 wells was analyzed after 3 days in vitro as described previously" (line 805), do you performed this experiments in technical and biological triplicate? Since it seems that the experiment was performed only once in technical triplicate. If it is not, please repeat the in vitro experiments.
Author response: We thank the reviewer for this comment. We preformed this experiment in biological triplicates and technical duplicates. The respective information has been updated in lines 846-847 of the revised manuscript.
Comment reviewer #2: Please add to Table 4 "Amplicon size (bp)" and "Amplification efficiency (%)"
Author response: We thank the reviewer for their suggestion. The expected amplicon sizes is provided in the revised Table 4. As we performed the qPCR with SYBR Green agent and not TaqMan, we are unfortunately not able to provide the amplification efficiency from our data. Additionally, the used instrument is not able to collect information about the amount of amplicon, which could be used to extrapolate the amplification efficiency. Please see: Line 848 in the revised manuscript.
Comment reviewer #2: 4.7. Statistical Analysis: please replace SEM with standard deviation in all your data Figures.
Author response: According to the reviewer’s request, we have revised all figures and tables. Please see: revised Figures 1, 2, 4, 5, and 8 and also the revised Tables 2 and 3 in the manuscript.
Round 2
Reviewer 2 Report
The authors responded adequately to my requests and improved the quality of their manuscript.
Nevertheless, I did not found the following sentences in the text, which should have been inserted in the text: “Group size was calculated by power analysis with Graphpad Statmate 2.0 software (GraphPad Software Inc., USA), based on data from previous investigation of another hydrogel (DOI: 10.3727/096368915X688010), with effect size >1, type I error: α: 0.05, p < 0.05, and type II error: β: 0.2, power 50%.” Please add it again.
Author Response
Authors’ point-by-point response to Reviewer #2:
Comment reviewer #2: The authors responded adequately to my requests and improved the quality of their manuscript.
Author response: We are very grateful to the reviewer for this positive comment.
Comment reviewer #2: Nevertheless, I did not found the following sentences in the text, which should have been inserted in the text: “Group size was calculated by power analysis with Graphpad Statmate 2.0 software (GraphPad Software Inc., USA), based on data from previous investigation of another hydrogel (DOI: 10.3727/096368915X688010), with effect size >1, type I error: α: 0.05, p < 0.05, and type II error: β: 0.2, power 50%.” Please add it again.
Author response: We deeply apologize for this sloppiness and thank the reviewer for their patience. Please find the information finally inserted to the section 4.2. Animals and surgery procedure lines 658-661 of the re-revised manuscript.
